# UNIFIED VISUAL TRANSFORMER COMPRESSION

**Shixing Yu**[1,*], **Tianlong Chen**[1,*], **Jiayi Shen**[2], **Huan Yuan**[3], **Jianchao Tan**[3],
**Sen Yang**[3], **Ji Liu**[3], **Zhangyang Wang**[1]
[1]University of Texas at Austin, [2]Texas A&M University, [3]Kwai Inc.
{shixingyu, tianlong.chen, atlaswang}@utexas.edu, asjyjya-617@tamu.edu,
{yuanhuan9412, senyang.nlpr, ji.liu.uwisc}@gmail.com, jianchaotan@kuaishou.com

## ABSTRACT

Vision transformers (ViTs) have gained popularity recently. Even without cus-
tomized image operators such as convolutions, ViTs can yield competitive perfor-
mance when properly trained on massive data. However, the computational over-
head of ViTs remains prohibitive, due to stacking multi-head self-attention modules
and else. Compared to the vast literature and prevailing success in compressing
convolutional neural networks, the study of Vision Transformer compression has
also just emerged, and existing works focused on one or two aspects of compres-
sion. This paper proposes a unified ViT compression framework that seamlessly
assembles three effective techniques: pruning, layer skipping, and knowledge dis-
tillation. We formulate a budget-constrained, end-to-end optimization framework,
targeting jointly learning model weights, layer-wise pruning ratios/masks, and
skip configurations, under a distillation loss. The optimization problem is then
solved using the primal-dual algorithm. Experiments are conducted with several
ViT variants, e.g. DeiT and T2T-ViT backbones on the ImageNet dataset, and our
approach consistently outperforms recent competitors. For example, DeiT-Tiny can
be trimmed down to 50% of the original FLOPs almost without losing accuracy.
Codes are available online: https://github.com/VITA-Group/UVC.

## 1 INTRODUCTION

Convolution neural networks (CNNs) (LeCun et al., 1989; Krizhevsky et al., 2012; He et al., 2016)
have been the *de facto* architecture choice for computer vision tasks in the past decade. Their training
and inference cost significant and ever-increasing computational resources. Recently, drawn by the
scaling success of attention-based models (Vaswani et al., 2017) in natural language processing (NLP)
such as BERT (Devlin et al., 2018), various works seek to leverage the Transformer architecture to
computer vision (Parmar et al., 2018; Child et al., 2019; Chen et al., 2020a). The Vision Transformer
(ViT) architecture (Dosovitskiy et al., 2020), and its variants, have been demonstrated to achieve
comparable or superior results on a series of image understanding tasks compared to the state of the
art CNNs, especially when pretrained on datasets with sufficient model capacity (Han et al., 2021).

Despite the emerging power of ViTs, such architecture is shown to be even more resource-intensive
than CNNs, making its deployment impractical under resource-limited scenarios. That is due to
the absence of customized image operators such as convolution, the stack of self-attention modules
that suffer from quadratic complexity with regard to the input size, among other factors. Owing to
the substantial architecture differences between CNNs and ViTs, although there is a large wealth
of successful CNN compression techniques (Liu et al., 2017; Li et al., 2016; He et al., 2017; 2019),
it is not immediately clear whether they are as effective as for ViTs. One further open question is
how to best integrate their power for ViT compression, as one often needs to jointly exploit multiple
compression means for CNNs (Mishra & Marr, 2018; Yang et al., 2020b; Zhao et al., 2020b).

On the other hand, the NLP literature has widely explored the compression of BERT (Ganesh et al.,
2020), ranging from unstructured pruning (Gordon et al., 2020; Guo et al., 2019), attention head
pruning (Michel et al., 2019) and encoder unit pruning (Fan et al., 2019); to knowledge distillation
(Sanh et al., 2019), layer factorization (Lan et al., 2019), quantization (Zhang et al., 2020; Bai et al.,

---

*Equal Contribution.

2020) and dynamic width/depth inference (Hou et al., 2020). Lately, earlier works on compressing ViTs have also drawn ideas from those similar aspects: examples include weight/attention pruning (Zhu et al., 2021; Chen et al., 2021b; Pan et al., 2021a), input feature (token) selection (Tang et al., 2021; Pan et al., 2021a), and knowledge distillation (Touvron et al., 2020; Jia et al., 2021). Yet up to our best knowledge, there has been no systematic study that strives to either compare or compose (even naively cascade) multiple individual compression techniques for ViTs – not to mention any joint optimization like (Mishra & Marr, 2018; Yang et al., 2020b; Zhao et al., 2020b) did for CNNs. We conjecture that may potentially impede more performance gains from ViT compression.

This paper aims to establish the first *all-in-one compression framework* that organically integrates three different compression strategies: (structured) pruning, block skipping, and knowledge distillation. Rather than ad-hoc composition, we propose a *unified vision transformer compression* (**UVC**) framework, which seamlessly integrates the three effective compression techniques and jointly optimizes towards the task utility goal under the budget constraints. UVC is mathematically formulated as a constrained optimization problem and solved using the primal-dual algorithm from end to end. Our main contributions are outlined as follows:

- We present UVC that unleashes the potential of ViT compression, by jointly leveraging multiple ViT compression means for the first time. UVC only requires to specify a global resource budget, and can automatically optimize the composition of different techniques.

- We formulate and solve UVC as a unified constrained optimization problem. It simultaneously learns model weights, layer-wise pruning ratios/masks, and skip configurations, under a distillation loss and an overall budget constraint.

- Extensive experiments are conducted with popular variants of ViT, including several DeiT backbones and T2T-ViT on ImageNet, and our proposal consistently performs better than or comparably with existing methods. For example, UVC on DeiT-Tiny (with/without distillation tokens) yields around 50% FLOPs reduction, with little performance degradation (only 0.3%/0.9% loss compared to the uncompressed baseline).

## 2 RELATED WORK

### 2.1 VISION TRANSFORMER

Transformer (Vaswani et al., 2017) architecture stems from natural language processing (NLP) applications first, with the renowned technique utilizing Self-Attention to exploit information from sequential data. Though intuitively the transformer model seems inept to the special inductive bias of space correlation for images-oriented tasks, it has proved itself of capability on vision tasks just as good as CNNs (Dosovitskiy et al., 2020). The main point of Vision Transformer is that they encode the images by partitioning them into sequences of patches, projecting them into token embeddings, and feeding them to transformer encoders (Dosovitskiy et al., 2020). ViT outperforms convolutional nets if given sufficient training data on various image classification benchmarks.

Since then, ViT has been developed to various different variants first on data efficiency towards training, like DeiT (Touvron et al., 2020) and T2T-ViT (Yuan et al., 2021) are proposed to enhance ViT's training data efficiency, by leveraging teacher-student and better-crafted architectures respectively. Then modifications are made to the general structure of ViT to tackle other popular downstream computer vision tasks, including object detection (Zheng et al., 2020; Carion et al., 2020; Dai et al., 2021; Zhu et al., 2020), semantic segmentation (Wang et al., 2021a;b), image enhancement (Chen et al., 2021a; Yang et al., 2020a), image generation (Jiang et al., 2021), video understanding (Bertasius et al., 2021), and 3D point cloud processing (Zhao et al., 2020a).

### 2.2 MODEL COMPRESSION

**Pruning.** Pruning methods can be broadly categorized into: unstructured pruning (Dong et al., 2017; Lee et al., 2018; Xiao et al., 2019) by removing insignificant weight via certain criteria; and structured pruning (Luo et al., 2017; He et al., 2017; 2018; Yu et al., 2018; Lin et al., 2018; Guo et al., 2021; Yu et al., 2021; Chen et al., 2021b; Shen et al., 2021) by zero out parameters in a structured group manner. Unstructured pruning can be magnitude-based (Han et al., 2015a;b), hessian-based (LeCun et al., 1990; Dong et al., 2017), and so on. They result in irregular sparsity, causing sparse matrix

operations that are hard to accelerate on hardware (Buluc & Gilbert, 2008; Gale et al., 2019). This can be addressed with structured pruning where algorithms usually calculate an importance score for some group of parameters (e.g., convolutional channels, or matrix rows). Liu et al. (2017) uses the scaling factor of the batch normalization layer as the sensitivity metric. Li et al. (2016) proposes channel-wise summation over weights as the metric. Lin et al. (2020) proposes to use channel rank as sensitivity metric while (He et al., 2019) uses the geometric median of the convolutional filters as pruning criteria. Particularly for transformer-based models, the basic structures that many consider pruning with include blocks, attention heads, and/or fully-connected matrix rows (Chen et al., 2020b; 2021b). For example, (Michel et al., 2019) canvasses the behavior of multiple attention heads and proposes an iterative algorithm to prune redundant heads. (Fan et al., 2019) prunes entire layers to extract shallow models at inference time.

**Knowledge Distillation.** Knowledge distillation (KD) is a special technique that does not explicitly compress the model from any dimension of the network. KD lets a student model leverage ··soft" labels coming from a teacher network (Hinton et al., 2015) to boost the performance of a student model. This can be regarded as a form of compression from the teacher model into a smaller student. The soft labels from the teacher are well known to be more informative than hard labels and leads to better student training (Yuan et al., 2020; Wei et al., 2020)

**Skip Configuration.** Skip connection plays a crucial role in transformers (Raghu et al., 2021), by tackling the vanishing gradient problem (Vaswani et al., 2017), or by preventing their outputs from degenerating exponentially quickly with respect to the network depth (Dong et al., 2021).

Meanwhile, transformer has an inborn advantage of a uniform block structure. A basic transformer block contains a Self-Attention module and a Multi-layer Perceptron module, and its output size matches the input size. That implies the possibility to manipulate the transformer depth by directly skipping certain layers or blocks. (Xu et al., 2020) proposes to randomly replace the original modules with their designed compact substitutes to train the compact modules to mimic the behavior of the original modules. (Zhang & He, 2020) designs a Switchable-Transformer Blocks to progressively drop layers from the architecture. To flexibly adjust the size and latency of transformers by selecting adaptive width and depth, DynaBERT (Hou et al., 2020) first trains a width-adaptive BERT and then allows for both adaptive width and depth. LayerDrop (Fan et al., 2019) randomly drops layers at training time; while at test time, it allows for sub-network selection to any desired depth.

## 3 METHODOLOGY

### 3.1 PRELIMINARY

**Vision Transformer (ViT) Architecture.** To unfold the unified algorithm in the following sections, here we first introduce the notations. There are totally $L$ transformer blocks. In each block $l$ of the ViT, there are two constituents, namely the Multi-head Self-Attention (MSA) module and the MLP module. Uniformly, each MSA for transformer block $l$ has $H$ attention heads originally.

In the Multi-head Self-Attention module, $W_Q^{(l)}$, $W_K^{(l)}$, $W_V^{(l)}$ are the weights of the three linear projection matrix in block $l$ that uses the block input $X^l$ to calculate attention matrices: $Q^l$, $K^l$, $V^l$. The weights of the projection module that follows self-attention calculation are denoted as $W^{(l,1)}$, represent the first linear projection module in block $l$. The MLP module consists of two linear projection modules $W^{(l,2)}$ and $W^{(l,3)}$.

**Compression Targets** The main parameters that can be potentially compressed in a ViT block are $W_Q^{(l)}$, $W_K^{(l)}$, $W_V^{(l)}$ and $W^{(l,1)}$, $W^{(l,2)}$, $W^{(l,3)}$. Our goal is to prune the head number and head dimensions simultaneously inside each layer, associated with the layer level skipping, solved in a unified framework. Currently, we do not extend the scope to reduce other dimensions such as input patch number or token size. However, our framework can also pack these parts together easily.

For head number and head dimensions pruning, instead of going into details of QKV computation, we innovate to use $\{W^{(l,1)}\}_{1 \le l \le L}$ to be the proxy pruning targets. Pruning on these linear layers is equivalent to the pruning of head number and head dimension. We also add $\{W^{(l,3)}\}_{1 \le l \le L}$ as our

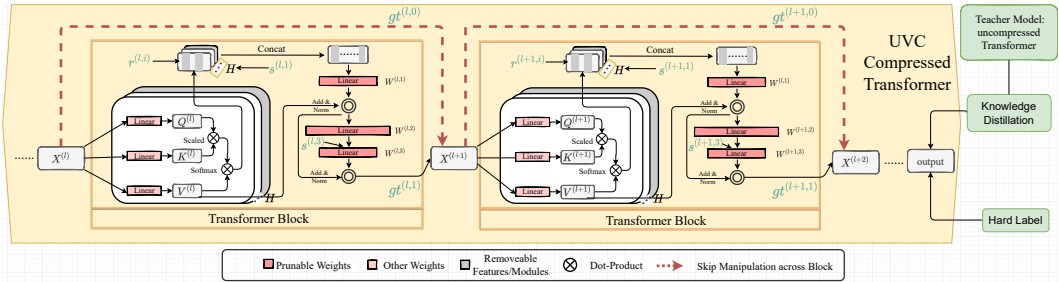

Figure 1: The overall framework of UVC, that integrates three compression strategies: (1) Pruning within a block: In a transformer block, we targeting on pruning Self-Attention head numbers ($s^{(l,1)}$), neuron numbers within a Self-Attention head ($r^{l,i}$) and the hidden size of MLP module ($s^{(l,3)}$) as well. (2) Skipping manipulation across blocks: When $\boldsymbol{gt}^{(l,0)}$ dominates, directly skip block $l$ and send $X^l$ into block $l+1$; Otherwise, pass $X^l$ into block $l$ without skip connection. (3) Knowledge distillation: The original uncompressed model is used to provide soft labels for knowledge distillation.

pruning targets, since these linear layers do not have dimension alignment issues with other parts, they can be freely pruned, while the output of $\{W^{(l,2)}\}$ should match with the dimension of block input. We do not prune $W_Q^{(l)}$, $W_K^{(l)}$, $W_V^{(l)}$ inside each head, since $Q^l$, $K^l$, $V^l$ should be of the same shape for computing self-attention.

Besides, skip connection is recognized as another important component to enhance the performance of ViTs (Raghu et al., 2021). The dimension between $X^l$ and the output of the linear projection module should be aligned. For this sake, $W^{(l,2)}$ is excluded from our compression. Eventually, the weights to be compressed in our subsequent framework are $\{W^{(l,1)}, W^{(l,3)}\}_{1 \leq l \leq L}$.

## 3.2 RESOURCE-CONSTRAINED END-TO-END VIT COMPRESSION

We target a principled constrained optimization framework jointly optimizing all weights and compression hyperparameters. We consider two strategies to compress ViT: (1) structural pruning of linear projection modules in a ViT block; and (2) adjusting the skipping patterns across different ViT blocks, including skipping/dropping an entire block. The second point, to our best knowledge, is the first time to be ever considered in ViT compression. The full framework is illustrated in Figure 1.

Alternatively, the two strategies could be considered as enforcing **mixed-level group sparsity**: the head dimension level, the head number level, and the block number level. The rationale is: when enforced with the same pruning ratios, models that are under finer-grained sparsity (i.e., pruning in smaller groups) are unfriendly to latency, while models that are under coarser-grained sparsity (i.e., pruning in larger groups) is unfriendly to accuracy. The mixed-level group sparsity can hence more flexibly trade-off between latency and accuracy.

The knowledge distillation is further incorporated into the objective to guide the end-to-end process. Below we walk through each component one-by-one, before presenting the unified optimization.

**Pruning within a Block**  To compress each linear projection module on width, we first denote $s^{(l,3)}$ as the number of columns to be pruned for weights $W^{(l,3)}$. Compression for weights $W^{(l,1)}$ is more complicated as it is decided by two degrees of freedom. As the input tensor to be multiplied with $W^{(l,1)}$ is the direct output of attention heads. Hence, within each layer $l$, we denote $s^{(l,1)}$ to be the attention head numbers that need to be pruned, and $r^{(l,i)}$ the number of output neurons to be pruned for each attention head $i$. Figure 2 illustrates the two sparsity levels: the head dimension level as controlled by $r^{(l,i)}$, and the head number level as controlled by $s^{(l,1)}$. They give more flexibility to transformer compression by selecting the optimal architecture in a multi-grained way. We emphasize again that those variables above are not manually picked, but rather optimized globally.

**Skipping Manipulation across Blocks**  The next component to jointly optimize is the skip connectivity pattern, across different ViT blocks. For vanilla ViTs, skip connection plays a crucial role in

ensuring gradient flow and avoiding feature collapse (Dong et al., 2021). Raghu et al. (2021) suggests that skip connections in ViT are even more influential than in ResNets, having strong effects on performance and representation similarity. However, few existing studies have systematically studied how to strategically adjust skip connections provided a ViT architecture, in order to optimize the accuracy and efficiency trade-off.

In general, it is known that more skip connections help CNNs' accuracy without increasing the parameter volume (Huang et al., 2017); and ViT seems to favor the same trend (Raghu et al., 2021). Moreover, adding a new skip connection over an existing block would allow us to "skip" it during inference, or "drop" it. This represents more aggressive "layer-wise" pruning, compared to the aforementioned element-wise or conventional structural pruning

While directly dropping blocks might look aggressive at the first glance, there are two unique enabling factors in ViTs that facilitate so. Firstly, unlike in CNNs, ViTs have uniform internal structures: by default, the input/output sizes of Self-Attention (SA) module, MLP module and thus the entire block, are all identical. That allows to painlessly drop any of those components and directly re-concatenating the others, without causing any feature dimension incompatibility. Secondly, Phang et al. (2021) observed that the top few blocks of fine-tuned transformers can be discarded without hurting performance, even with no further tuning. That is because deeper layer features tend to have high cross-layer similarities (while being dissimilar to early layer features), making additional blocks add little to the discriminative power. Zhou et al. (2021) also reported that in deep ViTs, the attention maps gradually become similar and even nearly identical after certain layers. Such cross-block "feature collapse" justifies our motivation to drop blocks.

In view of those, we introduce skip manipulation as a unique compression means to ViT for the first time. Specifically, for each transformer block with the skip connection, we denote $gt^{(l,0)}$ and $gt^{(l,1)}$ as two binary gating variables (summing up to 1), to decide whether to skip this block or not.

**The Constraints:** We next formulate weight sparsity constraints for the purpose of pruning. As discussed in Sec 3.1, the target of the proposed method is to prune the head number and head dimension simultaneously, which can actually be modeled as a two-level group sparsity problem when choosing $\{W^{(l,1)}\}_{1 \leq l \leq L}$ as proxy compression targets. Specifically, the input dimension of $\{W^{(l,1)}\}_{1 \leq l \leq L}$ is equivalent to the sum of the dimensions of all heads. Then we can put a two-level group sparsity regularization on $\{W^{(l,1)}\}_{1 \leq l \leq L}$ input dimension to compress head number and head dimension at the same time, as shown in 1a and 1b. $r^{(l,i)}$ corresponding to the pruned size of $i_{th}$ head. $s^{(l,1)}$ means the pruned number of heads. For $\{W^{(l,3)}\}_{1 \leq l \leq L}$, it is our compression target, we just perform standard one level group sparsity regularization on its input dimension, as shown in 1c.

$$\sum_j \mathbb{I}\left(\left\|\boldsymbol{W}_{g_{ij},.}^{(l,1)}\right\|_2^2 = 0\right) \geq r^{(l,i)}, \tag{1a}$$

$$\sum_i \mathbb{I}\left(\left\|\boldsymbol{W}_{g_{i},.}^{(l,1)}\right\|_2^2 = 0\right) \geq s^{(l,1)}, \tag{1b}$$

$$\sum_i \mathbb{I}\left(\left\|\boldsymbol{W}_{i,.}^{(l,3)}\right\|_2^2 = 0\right) \geq s^{(l,3)}, \tag{1c}$$

$$\forall\, l = 1, 2, ..., L, \forall\, i = 1, 2, ..., H \tag{1d}$$

where $\boldsymbol{W}_{i,.}$ denotes the $i$-th column of $\boldsymbol{W}$; $\boldsymbol{W}_{g_{i},.}$ denotes the $i$-th grouped column matrix of $\boldsymbol{W}$, i.e. the $i$-th head; and $\boldsymbol{W}_{g_{ij},.}$ the $j$-th column of $\boldsymbol{W}_{g_{i},.}$, which is the $j$-th column of the $i$-th head. In other words, in Eqn. 1b, column matrices are grouped by attention heads. Hence among the original $H$ heads in total at block $l$, at least $s^{(l,1)}$ heads should be discarded. Similarly, Eqn. 1c demands that at least $s^{(l,3)}$ input neurons should be pruned at this linear projection module. Furthermore, Eqn. 1a requests that in the $i$-th attention head at block $l$, $r(l,i)$ of the output units should be set to zeros.

Our method is formulated as a resource constrained compression framework, given a target resource budget, it will compress the model until the budget is reached. Given a backbone architecture, the FLOPs is the function of $s$, $r$ and $gt$, denoted as $\mathcal{R}_{\text{Flops}}(s, r, gt)$. We have the constraint as:

$$\mathcal{R}_{\text{Flops}}(\boldsymbol{s}, \boldsymbol{r}, \boldsymbol{gt}) \leq \mathcal{R}_{\text{budget}}, \tag{2}$$

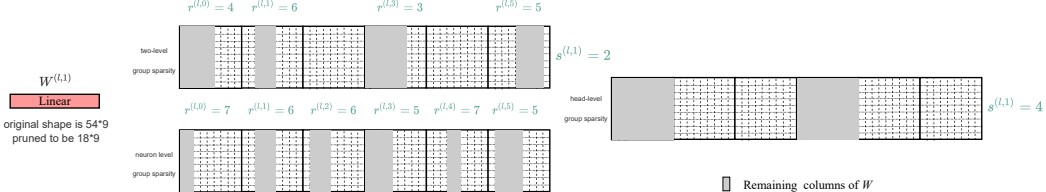

Figure 2: The two sparsity levels for pruning within a block: the head dimension level as controlled by $r^{(l,i)}$, and the head number level as controlled by $s^{(l,1)}$. When reaching the same pruning ratio, neuron-level sparsity will not remove any head, which is usually unfriendly to latency; while head level sparsity will only remove attention heads, which is usually unfriendly to accuracy.

where $\boldsymbol{s} = \{s^{(l,1)}, s^{(l,3)}\}_{1 \le l \le L}$ and $\boldsymbol{r} = \{r^{(l,i)}\}_{1 \le l \le L, 1 \le i \le H}$. $\mathcal{R}_{\text{budget}}$ is the resource budget. We present detailed flops computation equations in terms of $\boldsymbol{s}$, $\boldsymbol{r}$ and $\boldsymbol{gt}$ in Appendix.

Those inequalities can be further rewritten into equation forms to facilitate optimization. As an example, we follow (Tono et al., 2017) to reformulate Eqn. 1b as:

$$\sum_i \mathbb{I}\left(\|\boldsymbol{W}_{\cdot,g_i}\|_2^2 = 0\right) \ge s \Leftrightarrow \|\boldsymbol{W}_{\cdot,g}\|_{s,2}^2 = 0. \tag{3}$$

where $\|\boldsymbol{W}_{\cdot,g}\|_{s,2}^2$ denotes the Frobenius norm of the sub-matrix of $\boldsymbol{W}$ consisting of $s$ groups of $\boldsymbol{W}$ with smallest group norms. Eqn. 1a and Eqn. 1c have same conversions.

**The Objective:** The target objective could be written as ($\lambda$ is a coefficient):

$$\min_{\boldsymbol{W},\boldsymbol{gt}} \mathcal{L}(\boldsymbol{W}, \boldsymbol{gt}) = \ell(\boldsymbol{W}, \boldsymbol{gt}) + \lambda \ell_{distill}(\boldsymbol{W}, \boldsymbol{W}_t), \tag{4}$$

where $\boldsymbol{W}_t$ denotes the weights from teacher model, i.e., the uncompressed transformer model, and $\ell_{distill}(*, *)$ is defined as the **knowledge distillation** loss. We choose the simper $\ell_2$ norm as its implementation, as its performance was found to be comparable to the more traditional K-L divergence (Kim et al., 2021).

**The Final Unified Formulation:** Summarizing all above, we arrive at our unified optimization as a mini-max formulation, by leveraging the primal-dual method (Buchbinder & Naor, 2009):

$$\min_{\boldsymbol{W},\boldsymbol{s},\boldsymbol{r},\boldsymbol{gt}} \max_{\boldsymbol{p},\boldsymbol{y},z \ge 0} \mathcal{L}_{\text{pruning}} = \min_{\boldsymbol{W},\boldsymbol{s},\boldsymbol{r},\boldsymbol{gt}} \max_{\boldsymbol{p},\boldsymbol{y},z \ge 0} \mathcal{L}(\boldsymbol{W}, \boldsymbol{gt}) + \underbrace{z\left(R_{\text{Flops}}(\boldsymbol{s}, \boldsymbol{r}, \boldsymbol{gt}) - R_{\text{budget}}\right)}_{\text{resource loss}} +$$

$$\underbrace{\sum_{l=1}^{L}\left(y^{(l,1)}\left\|\boldsymbol{W}_{\cdot,g}^{(l,1)}\right\|_{\lceil s^{(l,1)}\rceil,2}^2 + y^{(l,3)}\left\|\boldsymbol{W}_{\cdot,\cdot}^{(l,3)}\right\|_{\lceil s^{(l,3)}\rceil,2}^2\right) + \sum_{l=1}^{L}\sum_{i=1}^{H} p^{(l,i)}\left\|\boldsymbol{W}_{\cdot,g_i\cdot}^{(l,1)}\right\|_{\lceil r^{(l,i)}\rceil,2}^2}_{\text{sparsity loss: } \boldsymbol{S}(\boldsymbol{y},\boldsymbol{s},\boldsymbol{p},\boldsymbol{r},\boldsymbol{gt},\boldsymbol{W})} \tag{5}$$

Eventually, we use the solution of the above mini-max problem, i.e. $\boldsymbol{s}$ and $\boldsymbol{r}$ to determine the compression ratio of each layer, and $\boldsymbol{gt}$ to determine the selection of skip configuration. Here, we select the pruned groups of the parameters by directly ranking columns by their norms, and remove those with the smallest magnitudes.

The general updating policy follows the idea of primal-dual algorithm. The full algorithm is outlined in Algorithm 1 in Appendix. We introduce the solutions to each sub-problem in Appendix as well.

## 4 EXPERIMENTAL RESULTS

**Datasets and Benchmarks** We conduct experiments for image classification on ImageNet (Krizhevsky et al., 2012). We implement UVC on DeiT (Touvron et al., 2020), which has basically the identical architecture compared with ViT (Dosovitskiy et al., 2020) except for an extra distillation token; and a variant of ViT – T2T-ViT (Yuan et al., 2021), with a special token embedding

layer that aggregates neighboring tokens into one token and a backbone with a deep-narrow structure. Experiment has been conducted on DeiT-Tiny/Small/Base and T2T-ViT-14 models. For DeiT-Tiny, we also try it without/with the distillation token. We measure all resource consumptions (including the UVC resource constraints) in terms of inference FLOPs.

**Training Settings**    The whole process of our method consists of two steps.

- *Step 1: UVC training*. We firstly conduct the primal-dual algorithm to the pretrained DeiT model to produce the compressed model under the given resource budget.
- *Step 2: Post training*. When we have the sparse model, we finetune it for another round of training to regain its accuracy loss during compression.

In the two steps mentioned above, we mainly follow the training settings of DeiT (Touvron et al., 2020) except for a relatively smaller learning rate which benefits finetuning of converged models.

For distillation, we select the pretrained uncompressed model to teach the pruned model.

Numerically, the learning rate for parameter $z$ is always changing during the primal-dual algorithm process. Thurs, we propose to use a dynamic learning rate for the parameter $z$ that controls the budget constraint. We use a four-step schedule of $\{1, 5, 9, 13, 17\}$ in practice.

**Baseline methods**    We adopt several latest compression methods which fall under two categories:

- *Category 1: Input Patch Reduction*, including $(i)$ **PoWER** (Goyal et al., 2020) : which accelerates language model BERT inference by eliminating word-vector in a progressive way. $(ii)$ **HVT** (Pan et al., 2021b): which reduce the token sequence dimension by using max-pooling hierarchically. $(iii)$ **PatchSlimming** (Tang et al., 2021): which identifies the effective patches in the last layer and then use them to guide the patch selection process of previous layers; and $(iv)$ **IA-RED$^2$** (Pan et al., 2021a): which dynamically and hierarchical drops visual tokens at different levels using a learned policy network.
- *Category 2: Model Weight Pruning*, including $(v)$ **SCOP** (Tang et al., 2020): which is originally a channel pruning method used on CNN, we follow (Tang et al., 2021) and implement it on ViT. $(vi)$ Vision Transformer Pruning (**VTP**) (Zhu et al., 2021): which trains transformers with sparsity regularization to let important dimensions emerge. $(vii)$ **SViTE** (Chen et al., 2021b): which jointly optimizes model parameters and explores sparse connectivity throughout training, ending up with one final sparse network. SViTE belongs to the most competitive accuracy-efficiency trade-off achieved so far, for ViT pruning. We choose its structured variant to be fair with UVC.

## 4.1 MAIN RESULTS

The main results are listed in  Tab. 1.  Firstly, we notice that most existing methods cannot save beyond 50% FLOPs without sacrificing too much accuracy.  In comparison, UVC can easily go with **larger compression rates** (up to $\geq 60\%$ FLOPS saving) without compromising as much. For example, when compressing DeiT-Tiny (with distillation token), UVC can trim the model down to a compelling $\geq 50\%$ of the original FLOPs while losing only 0.3% accuracy (Tab. 3). Especially, compared with the latest SViTE (Chen et al., 2021b) that can only save up to around 30% FLOPs, we observe UVC to **significantly outperform** ig at DeiT-Tiny/Small, at less accuracy drops (0.9/0.4, versus 2.1/0.6) with much more aggressive FLOPs savings (50.7%/42.4%, versus 23.7%/31.63). For the larger DeiT-Base, while UVC can save 45% of its FLOPs, we fin that SViTE cannot be stably trained at such high sparsity. UVC also yield comparable accuracy with the other ViT model pruning approach, VTP (Zhu et al., 2021), with more than 10% FLOPs savings w.r.t. the original backbone.

Secondly, UVC obtains strong results compared with Patch Reduction based methods. UVC performs clearly better than IA-RED$^2$ (Pan et al., 2021a) at DeiT-Base, and outperforms HVT (Pan et al., 2021b) on both DeiT-Tiny/Small models, which are latest and strong competitors. Compared to Patch Slimming Tang et al. (2021), UVC at DeiT-Small reaches 79.44% top-1 accuracy, while the compression ratio measured by FLOPs is comparable. On other models, we observe UVC to generally save more FLOPs, yet also sacrificing more accuracies. Moreover, as we explained in

Table 1: Comparison of the vision transformers compressed by UVC with different benchmarks on ImageNet. FLOPs remained denotes the remained ratio of FLOPs to the full-model FLOPs.

| Model | Method | Top-1 Acc. (%) | FLOPs(G) | FLOPs remained(%) |
|---|---|---|---|---|
| DeiT-Tiny | Baseline | 72.2 | 1.3 | 100 |
| | SViTE | 70.12 (-2.08) | 0.99 | 76.31 |
| | PatchSlimming | 72.0 (-0.2) | 0.7 | 53.8 |
| | UVC | 71.8 (-0.4) | 0.69 | 53.1 |
| | HVT | 69.7 (-2.5) | 0.64 | 49.23 |
| | UVC | 71.3 (-0.9) | 0.64 | 49.23 |
| | UVC | 70.6 (-1.6) | 0.51 | 39.12 |
| DeiT-Small | Baseline | 79.8 | 4.6 | 100 |
| | SViTE | 79.22 (-0.58) | 3.14 | 68.36 |
| | PoWER | 78.3 (-1.5) | 2.7 | 58.7 |
| | UVC | 79.44 (-0.36) | 2.65 | 57.61 |
| | SCOP | 77.5 (-2.3) | 2.6 | 56.4 |
| | PatchSlimming | 79.4 (-0.4) | 2.6 | 56.5 |
| | HVT | 78.0 (-1.8) | 2.40 | 52.2 |
| | UVC | 78.82 (-0.98) | 2.32 | 50.41 |
| DeiT-Base | Baseline | 81.8 | 17.6 | 100 |
| | IA-RED$^2$ | 80.3 (-1.5) | 11.8 | 67.04 |
| | SViTE | 82.22 (+0.42) | 11.87 | 66.87 |
| | VTP | 80.7 (-1.1) | 10.0 | 56.8 |
| | PatchSlimming | 81.5 (-0.3) | 9.8 | 55.7 |
| | UVC | 80.57 (-1.23) | 8.0 | 45.50 |
| T2T-ViT-14 | Baseline | 81.5 | 4.8 | 100 |
| | PoWER | 79.9 (-1.6) | 3.5 | 72.9 |
| | UVC | 80.4 (-1.1) | 2.90 | 60.4 |
| | UVC | 79.6 (-1.9) | 2.47 | 51.5 |
| | UVC | 78.9 (-2.6) | 2.11 | 44.0 |

Table 2: Ablation study on the modules implemented in UVC. The first part present single technique ablation. The second part presents the result to sequentially apply Skip configuration and Pruning.

| Method | DeiT-Tiny | |
|---|---|---|
| | Acc. (%) | FLOPs remained(%) |
| Uncompressed baseline | 72.2 | 100 |
| Only skip manipulation | 68.72 | 51.85 |
| Only pruning within a block | 70.52 | 50.69 |
| Without Knowledge Distillation | 69.34 | 51.23 |
| Skip→Prune 71.49%→49.39% | 66.84 | 49.39 |
| Prune→Skip 69.96%→54.88% | 69.68 | 54.88 |
| Prune→Skip 63.25%→50.73% | 70.02 | 50.73 |
| UVC | 71.3 | 49.23 |

Section 3.1, those input token reduction methods represent an orthogonal direction to the model weight sparsification way that UVC is pursuing. UVC can also be seamlessly extended to include token reduction into the joint optimization - a future work that we would pursue.

Thirdly, we test UVC on compressing T2T-ViT (Yuan et al., 2021). In the last row of Tab. 1, UVC with 44% FLOPs achieves only a 2.6% accuracy drop. Meanwhile, UVC achieves a 1.9% accuracy drop with 51.5% of the original FLOPs. With 60.4% FLOPs, UVC only suffers from a 1.1% accuracy drop, already outperforming PoWER with 72.9% FLOPs.

## 4.2 ABLATION STUDY

As UVC highlights the integration of three different compression techniques into one joint optimization framework, it would be natural to question whether each moving part is necessary for the pipeline, and how they each contribute to the final result. We conduct this ablation study, and the results are presented in Tab. 2.

**Skip Manipulation only.** We first present the result when we only conduct skip manipulation and knowledge distillation in our framework, but not pruning within a block. All other protocols remain unchanged. This two-in-a-way method is actually reduced into LayerDropping (Lin et al., 2018). We have the following findings: Firstly, implementing only skip connection manipulation will incur high instability. During optimization, the objective value fluctuates heavily due to the large architecture changes (adding or removing one whole block) and barely converges to a stable solution. Secondly, only applying skip manipulation will be damaging the accuracy remarkably, e.g., by nearly 4% drop on DeiT-Tiny. That is expected, as only manipulating the model architecture with such a coarse granularity (keeping or skipping a whole transformer block) is very inflexible and prone to collapse.

**Pruning within a Block only.** We then conduct experiments without using the gating control for skip connections. That leads to only integrating neuron-level and attention-head-level pruning, with knowledge distillation. It turns out to deliver much better accuracy than only skip manipulation, presumably owing to its finer-grained operation. However, it still has a margin behind our joint UVC method, as the latter also benefits from removing block-level redundancy *a priori*, which is recently observed to widely exist in fine-tuned transformers Phang et al. (2021). Overall, the ablation results endorse the effectiveness of jointly optimizing those components altogether.

**Applying Individual Compression Methods Sequentially** Our method is a joint optimization process for skip configuration manipulation and pruning, but is this joint optimization necessary?

To address this curiosity, we mark Prune→Skip as the process of pruning first, manipulating skip configuration follows, while Skip→Prune vice versa. Note that Skip→Prune 71.49%→49.39% denotes the pipeline that use skip configuration first to enforce the model FLOPs to be 71.49% of the original FLOPs, then apply pruning to further compress the model to 49.39%. Without loss of generality, we approximate both the pruning procedure and skipping procedure to compress 70% of the previous model, which will approximately result in a model with 50% of the original FLOPs. Based on this, we implement:$(i)$ **Skip→Prune 71.49%→49.39%** It first compresses the model to 71% by skipping then applies pruning on the original model and results in the worse performance at 66.84%. $(ii)$**Prune→Skip 69.96%→54.88%** It applies pruning to 70% FLOPs then skipping follows. We observe that applying pruning first can stabilize the choice of skip configuration and result in a top-1 accuracy of 69.68%. Both results endorse the superior trade-off found by UVC.

Furthermore, we design a better setting (Prune→Skip 63.25%→50.73%) to simulate the compression ratio found by UVC. Details can be found in Appendix. The obtained result is much better than previous ones, but still at a 1% performance gap from the result provided by UVC. We assign the extra credit to jointly optimizing the architecture and model weights.

## 5 CONCLUSION

In this paper, we propose UVC, a unified ViT compression framework that seamlessly assembles pruning, layer skipping, and knowledge distillation as one. A budget-constrained optimization framework is formulated for joint learning. Experiments demonstrate that UVC can aggressively trim down the prohibit computational costs in an end-to-end way. Our future work will extend UVC to incorporating weight quantization as part of the end-to-end optimization as well.

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

# A APPENDIX

## A.1 UPDATING POLICY

**Updating Weights $W$** Different from other pruning methods in CNN that use fixed pre-trained weights to select the pruned channels/groups with certain pre-defined metrics (Lin et al., 2020; He et al., 2019), here our subproblem could be considered as following dynamic pruning criteria that will be updated along. Specifically, we solve the following subproblem:

$$\text{Prox}_{\eta_1 S(\boldsymbol{y}, \boldsymbol{s}, \boldsymbol{p}, \boldsymbol{r}, \boldsymbol{W})}(\bar{\boldsymbol{W}}) = \arg\min_{\boldsymbol{W}} \frac{1}{2} \left\| \boldsymbol{W} - \bar{\boldsymbol{W}} \right\|^2 + \eta_1 \boldsymbol{S}(\boldsymbol{y}, \boldsymbol{s}, \boldsymbol{p}, \boldsymbol{r}, \boldsymbol{W}), \tag{6}$$

where $\bar{\boldsymbol{W}} = \boldsymbol{W}^t - \eta_1 \hat{\nabla}_{\boldsymbol{W}} \ell(\boldsymbol{W}^t)$. The solution admits a bi-level projection (Yang et al., 2016):

$$\boldsymbol{W}^{(l,1)*}_{.,g_{ij}} = \begin{cases} \bar{\boldsymbol{W}}^{(l,1)}_{.,g_{ij}}, & \text{if } \left\| \bar{\boldsymbol{W}}^{(l,1)}_{.,g_{ij}} \right\|^2_2 \geq \left\| \boldsymbol{W}^{(l,1)}_{.,g_{i\text{least-}\lceil r^{(l,i)} \rceil}} \right\|^2_2, \\ \frac{1}{1+2\eta_1 p^{(l,i)}} \bar{\boldsymbol{W}}_i, & \text{otherwise,} \end{cases}$$

$$\boldsymbol{W}^{(l,1)*}_{.,g_i} = \begin{cases} \bar{\boldsymbol{W}}^{(l,1)}_{.,g_i}, & \text{if } \left\| \bar{\boldsymbol{W}}^{(l,1)}_{.,g_i} \right\|^2_2 \geq \left\| \boldsymbol{W}^{(l,1)}_{.,g_{\text{least-}\lceil s(l,1) \rceil}} \right\|^2_2, \\ \frac{1}{1+2\eta_1 y^{(l,1)}} \bar{\boldsymbol{W}}_{.,g_i}, & \text{otherwise,} \end{cases}$$

$$\boldsymbol{W}^{(l,3)*}_{.,i} = \begin{cases} \bar{\boldsymbol{W}}^{(l,3)}_{.,i}, & \text{if } \left\| \bar{\boldsymbol{W}}^{(l,3)}_{.,i} \right\|^2_2 \geq \left\| \boldsymbol{W}^{(l,3)}_{.,\text{least-}\lceil s(l,3) \rceil} \right\|^2_2, \\ \frac{1}{1+2\eta_1 y^{(l,3)}} \bar{\boldsymbol{W}}_{.,i}, & \text{otherwise,} \end{cases}$$

where least-$j$ denotes the index of the (group) columns of $\boldsymbol{W}$ that have $j$-th least (group) norm.

**Updating $\boldsymbol{gt}$** $\boldsymbol{gt}^l = (\boldsymbol{gt}^{(l,0)}, \boldsymbol{gt}^{(l,1)})$ are used to generate a binomial categorical distribution to decide whether pass through block $l$ or directly skip it. As the two variables are discrete, we apply the renowned `Gumbel-Softmax` (GSM) trick (Jang et al., 2016) to obtain differentiable and polarized sampling. For $\boldsymbol{gt}^l = (\boldsymbol{gt}^{(l,0)}, \boldsymbol{gt}^{(l,1)})$, given i.i.d Gumbel noise $g$ drawn from $Gumbel(0,1)$ distribution, a soft categorical sample can be drawn by

$$G^l = GSM(\boldsymbol{gt}^l) = Softmax((log(\boldsymbol{gt}^l) + g)/\tau) \in \mathbb{R}^2, \tag{7}$$

where $G^{(l,1)}$ refers to the continuous possibility to preserve current block $l$ while $G^{(l,0)}$ to drop it. Directly applying the chain rule on $\mathcal{L}(\boldsymbol{W}, \boldsymbol{gt})$ w.r.t $\boldsymbol{gt}$ can now calculate $\tilde{\nabla}_{\boldsymbol{gt}} \mathcal{L}(\boldsymbol{W}^t, \boldsymbol{gt}^t)$.

Also, $\boldsymbol{gt}$ participates in the calculation of FLOPs, by deciding whether to pass certain blocks. Since passing or skipping a block is a dynamic choice during training, we estimate the FLOPs of the $l$-th block $\mathcal{R}_{\text{Flops}}(\boldsymbol{s}, \boldsymbol{r}, \boldsymbol{gt})$ with skip gating by using its expectation. To be specific,

$$\mathcal{R}_{\text{Flops}_l}(\boldsymbol{s}_l, \boldsymbol{r}_l, \boldsymbol{gt}_l) = \mathbb{E}[\mathcal{R}_{\text{Flops}_l}(\boldsymbol{s}_l, \boldsymbol{r}_l, \boldsymbol{gt}_l) | \boldsymbol{s}_l, \boldsymbol{r}_l] \tag{8a}$$

$$= G^{(l,0)} \mathcal{R}_{\text{Flops}_l}(Identity) + G^{(l,1)} \mathcal{R}_{\text{Flops}_l}(\boldsymbol{s}_l, \boldsymbol{r}_l) \tag{8b}$$

$$= G^{(l,1)} \mathcal{R}_{\text{Flops}_l}(\boldsymbol{s}_l, \boldsymbol{r}_l) \tag{8c}$$

Hence, updating policy for $\boldsymbol{gt}$ is formulated as:

$$\boldsymbol{gt}^{t+1} = \boldsymbol{gt}^t - \eta_4 \left( \tilde{\nabla}_{\boldsymbol{gt}} \mathcal{L}(\boldsymbol{W}, \boldsymbol{gt}^t) + \tilde{\nabla}_{\boldsymbol{gt}} z \left( R_{\text{Flops}}(\boldsymbol{s}, \boldsymbol{r}, \boldsymbol{gt}^t) - R_{\text{budget}} \right) \right) \tag{9}$$

$$= \boldsymbol{gt}^t - \eta_4 \left( \tilde{\nabla}_{\boldsymbol{gt}} \mathcal{L}(\boldsymbol{W}, \boldsymbol{gt}^t) + \tilde{\nabla}_{\boldsymbol{gt}} z R_{\text{Flops}}(\boldsymbol{s}, \boldsymbol{r}) GSM(\boldsymbol{gt}^t) \right) \tag{10}$$

**Updating $s$ and $r$** Similar to the updating policy of $gt$, one gradient term w.r.t. $s$ and $r$ are $\tilde{\nabla}_s z \left( R_{\text{Flops}} \left( s, r, gt \right) - R_{\text{budget}} \right)$, $\tilde{\nabla}_r z \left( R_{\text{Flops}} \left( s, r, gt \right) - R_{\text{budget}} \right)$ respectively.

The other gradient term is calculated on the unified formulation Eqn. 5. Refer to that, $s$ and $r$ are floating-point numbers during the optimization process. In practise, ceiling functions are operated on them to determine the integer number that should be pruned for each layer. However, the ceiling function $\lceil . \rceil$ is non-differentiable. To solve this problem, we implement Straight-through estimator(STE) (Bengio et al., 2013) to provide a proxy of the gradient when performing the backward pass. We set $\frac{\partial \lceil s \rceil}{\partial s} = 1$.

As for $\left\| W_{.,g} \right\|_{s,2}^2$ term in the sparsity loss, we use $\left\| W_{.,g} \right\|_{s+1,2}^2 - \left\| W_{.,g} \right\|_{s,2}^2$ as the proxy of partial derivative of $\left\| W_{.,g} \right\|_{s,2}^2$ with respect to $s$:

$$\frac{\tilde{\partial} \left\| W_{.,g} \right\|_{s,2}^2}{\tilde{\partial} s} = \left\| W_{.,g_{\text{least-min}\{\text{Dim}(W), s+1\}}} \right\|_2^2, \tag{11}$$

where $\text{Dim}(W)$ is the number of column groups of $W$. The other two terms in the sparsity loss can be processed similarly.

## A.2 MAIN ALGORITHM

The general updating policy follows the idea of primal-dual algorithm. The full algorithm is outlined in Algorithm 1.

---

**Algorithm 1:** Gradient-based algorithm to solve problem (5) for Unified ViT Compression.

---
**Input:** Resource budget $R_{\text{budget}}$, learning rates $\eta_1, \eta_2, \eta_3, \eta_4, \eta_5, \eta_6$, number of total iterations $\tau$.
**Result:** Transformer pruned weights $W^*$.

1 Initialize $t = 1, W^1$ ;                           // random or a pre-trained dense model
2 **for** $t \leftarrow 1$ **to** $\tau$ **do**

3      $W^{t+1} = \text{Prox}_{\eta_1 S(y^t, s^t, p^t, r^t, W^t)} \left( W^t - \eta_1 \hat{\nabla}_W \mathcal{L} \left( W^t, gt \right) \right)$ ;           // Proximal-SGD

4      $s^{t+1} = s^t - \eta_2 \left( \tilde{\nabla}_s S \left( y^t, s^t, p^t, r^t, gt^t, W^{t+1} \right) + \tilde{\nabla}_s z^t \left( R_{\text{Flops}} \left( s^t, r^t, gt^t \right) - R_{\text{budget}} \right) \right)$ ;
     // Gradient (STE) Descent

5      $r^{t+1} = r^t - \eta_3 \left( \tilde{\nabla}_r S \left( y^t, s^{t+1}, p^t, r^t, gt^t, W^{t+1} \right) + \tilde{\nabla}_r z^t \left( R_{\text{Flops}} \left( s^{t+1}, r^t, gt^t \right) - R_{\text{budget}} \right) \right)$ ;
     // Gradient (STE) Descent

6      $gt^{t+1} = gt^t - \eta_4 \left( \tilde{\nabla}_{gt} \mathcal{L} \left( W^{t+1}, gt^t \right) + \tilde{\nabla}_{gt} z^t \left( R_{\text{Flops}} \left( s^{t+1}, r^{t+1}, gt^t \right) - R_{\text{budget}} \right) \right)$ ;
     // Gradient Descent

7      $z^{t+1} = z^t + \eta_7 \left( R_{\text{Flops}} \left( s^{t+1}, r^{t+1}, gt^{t+1} \right) - R_{\text{budget}} \right)$ ;           // Gradient Ascent

8      $y^{(l,1)t+1} = y^{(l,1)t} + \eta_5 \left( \left\| W_{.,g}^{(l,1)t+1} \right\|_{\lceil s^{(l,1)t+1} \rceil, 2}^2 \right)$ ;           // Gradient Ascent

9      $y^{(l,3)t+1} = y^{(l,3)t} + \eta_5 \left( \left\| W_{.,.}^{(l,3)t+1} \right\|_{\lceil s^{(l,3)t+1} \rceil, 2}^2 \right), \forall l = 1, \cdots, L$

     $p^{(l,i)t+1} = p^{(l,i)t} + \eta_6 \left( \left\| W_{.,g_i.}^{(l,1)t+1} \right\|_{\lceil r^{(l,i)t+1} \rceil, 2}^2 \right), \forall i = 1, \cdots, H, \forall l = 1, \cdots, L$

10 $W^* = W$

---

## A.3 VISUALIZATION

To reveal more intuitions how UVC optimize the architecture, we present two groups of visualizations:

- **Sparse connectivity patterns:** We first visualize the sparse connectivity patterns in Fig. 3 (a), i.e, how many attention heads are preserved, and how they (sparsely) distribute over all blocks. For the smaller DeiT-Tiny model, UVC tends to more evenly drop attention heads across layers; meanwhile, for the larger DeiT-Base, UVC clearly prefers to drop more attention heads in the early layers.

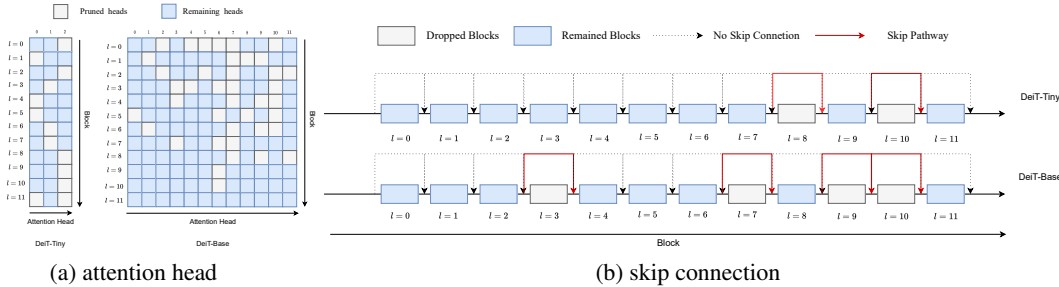

(a) attention head                        (b) skip connection

Figure 3: Visualizations of: (a) sparse attention head patterns for DeiT-Tiny (left) and DeiT-Base (right); (b) skip connection patterns for DeiT-Tiny (top) and DeiT-Base (bottom).

Table 3: Result with distillation token of DeiT

| Model | Method | Top-1 Acc. (%) | FLOPs(G) | FLOPs remained(%) |
|---|---|---|---|---|
| DeiT-Tiny | Baseline | 74.4 | 1.3 | 100 |
| (dist token) | UVC | 74.1(-0.3) | 0.66 | 50.58 |

- **Skip connection patterns:** We next draw the result of skip connection patterns Fig. 3 (b), showing what layers are learned to be directly skipped under unified optimization. We observe UVC's obvious tendency to drop later layers, which coincides nicely with the observations by (Phang et al., 2021; Zhou et al., 2021).

As is observed in Fig. 3, DeiT-Tiny discarded 2 blocks in total, which results in 84% of compression ratio. Thus, in Sec. 4.2, to design an optimal architecture for sequentially applying compression methods, we first prune the DeiT-Tiny to 63% and apply skipping to indulge a 50% FLOPs' model to mimick the behavior of UVC.

## A.4 EXTRA RESULTS

In this section, we present the result of DeiT with distillation token to extract the best performance vision transformer can reach under UVC framework. Results are shown in 3.

