# OpenReview forum: "Unified Visual Transformer Compression"
_ICLR.cc/2022/Conference — ICLR 2022 Poster_

### Official Review · Reviewer_K4Gq · 2021-10-31

**Correctness:** 4
**Technical Novelty And Significance:** 3
**Empirical Novelty And Significance:** 4
**Recommendation:** 6
**Confidence:** 4

**Main Review:**

The authors’ efforts to formulate and solve ViT compression as an end-to-end budget-constrained optimization is appreciated. Directly skipping layers is a fresh idea as a compression means and seems particularly to fit ViT due to its homogenous block structure and common feature collapse. This makes a brand-new angle to compress ViTs. The formulation and algorithm are correct to my best knowledge. UVC only requires specifying an overall resource budget and can automatically optimize the composition of different techniques. It is easier to use than ad-hoc alternatives while being very flexible. UVC experimentally outperformed a few very recent baselines (e.g, surpassing SViTE of NeurIPS’21), accompanied by sufficient empirical dive-in with ablation studies and visualization.

The novelty is okay but not so great. Combining two or three compression means is well studied in CNN compression. For UVC’s specific combination, the distillation loss is a rather straightforward plug-in, and isn’t the same technically interesting as the other two in joint optimization.
The performance isn’t clearly better than Patch Slimming, mostly due to the current mismatched FLOPs. If the authors cannot show better than it in a fair setting, then I suggest the authors to demonstrate they can be combined so UVC’s empirical benefits can still stand.
A major caveat in experiments is that the authors never reported their due ablation experiments. A clearly missing baseline is: what if you apply three compression methods sequentially, such as first pruning then skipping layer then distilling (or in whatever other reasonable orders)? This missing is crucial as no existing experiment can directly support joint optimization is essential
There is no DeiT-small in table and I don’t understand why. The compression ratios chosen to display are very “sparse” (kinda understandable, since training VITs is heavy). How will UVC perform if increasing compression ratios further?
(Optional) it would further strengthen the paper if more strong ViT variants can be reported such as Swin Transformer or T2T.

---
The authors have well addressed my major concerns. I am glad to increase my score to 6.

**Summary Of The Paper:**

This is an interesting work trying to assemble three effective techniques for pruning, layer skipping, and knowledge distillation.

**Summary Of The Review:**

While this joint optimization is a reasonably novel effort in the (somehow niche) ViT compression field, I cannot endorse this paper further given several critical experiments are missing. I’m willing to revisit my rating if the authors can provide the above questioned results.

---

> ### Author Response · Authors · 2021-11-20
> **Response to Reviewer K4Gq (Cons 1&2)**
>
> Thanks for rating our paper as interesting and meaningful. We provide pointwise responses to your concerns about our experiment settings below.
>
> **[Cons1. Combining two or three compression means is well studied]**
>
> We respectfully argue that the whole framework is clearly not a simple combination of different techniques, and has never been explored in the efficient vision transformer domain. In the “new sub-field of ViT compression” (quoting Reviewer **UdwX**), our main innovations include:
> (1) the solid, mathematically well-grounded joint optimization form, quoting **UdwX**: “the rigorous optimization in this paper. Different from previous ad-hoc methods that need tune more hyperparameters, UVC can compress with one global budget control, and jointly coordinate different compression forms automatically under the hood”, and Reviewer **K4Gq** comment that “the efforts to formulate and solve ViT compression as an end-to-end budget-constrained optimization is appreciated”;
> (2) the novel mixed-level group sparsity formulated for ViT compression, and particularly (quoting **UdwX**) “ including layer-skipping as a compression means is novel, and to my best knowledge not considered by previous joint compression works yet.”, and (As you have mentioned) “Directly skipping layers is a fresh idea as a compression mean and seems particular to fit ViT due to its homogenous block structure and common feature collapse. This makes a brand-new angle to compress ViTs.”
>
>
> **[Cons2. Performance isn’t clearly better than Patch Slimming]**
>
> Thanks for your suggestion. It is worth mentioning that our proposed frameworks can integrate many other compression mechanisms into the same framework. Especially what the reviewer mentioned here with PatchSlimming, which is the token-level compression that can massively reduce the computation cost by calculating a smaller attention map. Since the code of Patch Slimming is provisionally not publicly available, we implemented a token selection technique using the code base of SViTE[1] (Specifically, SViTE+) into part of UVC, during the rebuttal window, and the model is currently running with promising intermedia results. We test this UVC-s (short for “UVC augmented by SVITE+ patch reduction“) on the DeiT-Tiny model and the result is listed in the following Table R1:
>
> **Table R1**
>
> |Method  	|         Acc  	| FLOPs(G) | Compression Ratio (%)|
> |:-:|:-:|:-:|:-:|
> |DeiT-Tiny |        72.2		|     	1.3    | 100|
> |UVC	|    71.3 (-0.9)  	| 	0.64 	| 49.23|
> |UVC-s	|    71.5 (-0.7) 	| 	0.65 	| 50.12|
>
> The UVC-s framework can include Patch Slimming work into our primal-dual optimization, instead of deriving their original greedy method.  And UVC-s can perform better than UVC-only. This result proves that Patch Slimming is orthogonal to UVC and adding it to our framework can further boost the performance.
>
> [1] Tianlong Chen, Yu Cheng, Zhe Gan, Lu Yuan, Lei Zhang, and Zhangyang Wang. Chasing sparsity in vision transformers: An end-to-end exploration. arXiv preprint arXiv:2106.04533, 2021b.

---

> ### Author Response · Authors · 2021-11-20
> **Response to Reviewer K4Gq (Cons 3)**
>
> **[Cons3. Apply compression methods sequentially]**
>
> Thank you for pointing out the important baselines of sequentially applying the compression techniques. We provide three new ablation settings to study the result of sequentially applying compression techniques. We mark Prune->Skip as the process of first pruning then manipulating skip configuration, while Skip->Prune vice versa.
>
> **Table R2**
>
> | Method | Uncompressed DeiT-Tiny | Skip->Prune 71.49%->49.39% | Prune->Skip 69.96%->54.88% | Prune->Skip 63.25%->50.73% | UVC |
> |:-:|:-:|:-:|:-:|:-:|:-:|
> | Acc. (%)| 72.2 | 66.84 | 69.68 | 70.02 | 71.3 |
> | FLOPs(%)| 100 | 49.39 | 54.88 | 50.73 | 49.23 |
>
> Without loss of generality, we approximate both the pruning procedure and skipping procedure to compress 70% of the previous model, which will approximately result in a model with 50% of the original FLOPs. Based on this, we implement:
>
> - (Skip->Prune 71.49%->49.39%) It first compresses the model to 71% by skipping then applies pruning on the original model and results in the worse performance at 66.84%.
> - (Prune->Skip 69.96%->54.88%) It applies pruning to 70% FLOPs then skipping follows.
>
> We observe that applying pruning first can stabilize the choice of skip configuration and result in a top-1 accuracy of 69.68%. Both of the results show that UVC finds the best trade-off between skip and pruning.
>
> Furthermore, we design a better setting (Prune->Skip 63.25%->50.73%) to simulate the compression ratio found by UVC, as is observed in Figure 3b in the main text, DeiT-Tiny discarded 2 blocks in total, which results in 84% of compression ratio. Thus, we first prune the DeiT-Tiny to 63% and apply skip to indulge a 50% FLOPs’ model.
>
> The obtained result of the setting (Prune->Skip 63.25%->50.73%)  is much better than previous ones with a sub-optimal trade-off, but still has a 1% performance gap with the result provided by UVC. We give the credit to jointly optimize the architecture and model weights to avoid local-minima convergence for architecture.

---

> ### Author Response · Authors · 2021-11-20
> **Response to Reviewer K4Gq (Cons 4)**
>
> **[Cons4. No DeiT-small in table. How will UVC perform if increasing compression ratios further]**
>
> *<Why no DeiT-Small?>*
>
> Thanks. To address your concern, we present new results of DeiT-Small in Table R3 below, where the comparison with **Patch Slimming** is included. Note that due to the limited rebuttal time window, the experiments are still on the run, and UVC is still evolving.
>
> **Table R3**
>
> |Method  	|         Acc  	| FLOPs(G) | Compression Ratio (%)|
> |:-:|:-:|:-:|:-:|
> |DeiT-Small 		|        79.8		|     	4.6    | 100|
> |SCOP[1] 			|    77.5 (-2.3) 	| 	2.6 	| 56.4|
> |PoWER[2] 		|    78.3 (-1.5) 	| 	2.7 	| 58.7|
> |HVT [3] 			|    78.0 (-1.8)  	| 	2.4 	| 52.2|
> |Patch Slimming [4] 	|    79.4 (-0.4)  	| 	2.6	| 56.5|
> |UVC (Ours)		|    79.44 (-0.36)| 	2.65 	| 57.61|
> |UVC (Ours)		|    78.82 (-0.98)| 	2.32 	| 50.41|
>
> As seen from Table R3, UVC already outperformed recent strong competitors such as HVT (published in ICCV’21) and others. The newly updated result has a comparable top-1 accuracy of 79.44% to Patch Slimming, while the compression ratio measured by FLOPs is similar to what Patch Slimming obtains. We post this intermediate - non-final and not conclusive - result here to ensure timely feedback, but will also update newer results once we have it. We publicly promise that DeiT-small compression results will be reported in the final draft.
>
> *<If increasing compression ratios further?>*
>
> Thanks for your understanding about the heavy ViT training. We investigate more aggressive compression ratios on DeiT-Tiny, as shown in the below table.
>
> **Table R4**
>
> |Method  	|         Acc  	| FLOPs(G) | Compression Ratio (%)|
> |:-:|:-:|:-:|:-:|
> |DeiT-Tiny 	|        72.2		|     	1.3    | 100|
> |UVC (High)	|    71.8 (-0.4) 	| 	0.7 	| 53.1|
> |UVC (Medium)	|    71.3 (-0.9)  	| 	0.64 	| 49.23|
> |UVC (Low)	|    70.6 (-1.6) 	| 	0.5 	| 39.12|
>
> All the above results are obtained from the DeiT-Tiny model, and with a high compression ratio of 39.12% FLOPs remained (Hardly reported in the literature on ViT), the compressed model from UVC still maintains reasonable accuracy of 70.6%.
>
> [1] Yehui Tang, Yunhe Wang, Yixing Xu, Dacheng Tao, Chunjing XU, Chao Xu, and Chang Xu. Scop: Scientific control for reliable neural network pruning. In H. Larochelle, M. Ranzato, R. Hadsell, M. F. Balcan, and H. Lin, editors, Advances in Neural Information Processing Systems, volume 33, pages 10936–10947. Curran Associates, Inc., 2020.
>
> [2] Saurabh Goyal, Anamitra Roy Choudhury, Saurabh Raje, Venkatesan Chakaravarthy, Yogish Sabharwal, and Ashish Verma. Power-bert: Accelerating bert inference via progressive word-vector elimination. In International Conference on Machine Learning, pages 3690–3699. PMLR, 2020.
>
> [3] Pan, Zizheng and Zhuang, Bohan and Liu, Jing and He, Haoyu and Cai, Jianfei.Scalable Vision Transformers With Hierarchical Pooling. In Proceedings of the IEEE/CVF International Conference on Computer Vision (ICCV), pages 377-386, 2021.
>
> [4] Yehui Tang, Kai Han, Yunhe Wang, Chang Xu, Jianyuan Guo, Chao Xu, and Dacheng Tao. Patch slimming for efficient vision transformers. arXiv preprint arXiv:2106.02852, 2021.

---

> ### Author Response · Authors · 2021-11-25
> **Response to Reviewer K4Gq (Cons 5)**
>
> **[Cons5. Strong ViT variants such as Swin Transformer or T2T performance]**
>
> Thanks for the suggestions. We have applied UVC on the backbone of T2T-ViT-14 to test the effectiveness of our proposals across different architectures. Since T2T-ViT-14 has much larger computational costs and is slower to train/run, during this limited rebuttal time and resource, as of now, the updated results on T2T-ViT-14 on diverse ratios are listed below:
>
> **Table R5**
>
> |Method  	|         Acc  	| FLOPs(G) | Compression Ratio (%)|
> |:-:|:-:|:-:|:-:|
> |T2T-ViT-14 		|        81.5		|     	4.8    | 100	|
> |PoWER[1] 		|    79.9(-1.6)	| 	3.5 	| 72.9|
> |UVC (high)		|    80.4 (-1.1) 	| 	2.90 	| 60.4|
> |UVC (medium)		|    79.6 (-1.9) 	| 	2.47 	| 51.5|
> |UVC (low)			|    78.9 (-2.6) 	| 	2.11 	| 44.0|
>
> As is shown in Table R5, UVC with 44% FLOPs achieves only a 2.6% accuracy drop. Meanwhile, UVC achieves a 1.9% accuracy drop with 51.5% of the original FLOPs. These two results are quite significant with a much larger compression ratio than PoWER performs. While with 60.4% FLOPs, UVC only suffers from a 1.1% accuracy drop, significantly outperforming PoWER with 72.9% FLOPs. This provides evidence that our method is independent of backbone and works well on both DeiT and T2T-ViT. We post this intermediate - non-final and not conclusive - result here to ensure timely feedback, but will also update newer results once we have it. We publicly promise that the T2T-ViT-14 compression results will be reported in the final draft.
>
> [1] Saurabh Goyal, Anamitra Roy Choudhury, Saurabh Raje, Venkatesan Chakaravarthy, Yogish Sabharwal, and Ashish Verma. Power-bert: Accelerating bert inference via progressive word-vector elimination. In International Conference on Machine Learning, pages 3690–3699. PMLR, 2020.

---

### Official Review · Reviewer_UdwX · 2021-11-01

**Correctness:** 4
**Technical Novelty And Significance:** 3
**Empirical Novelty And Significance:** 4
**Recommendation:** 8
**Confidence:** 4

**Main Review:**

Motivation and Overall Structure:
While a few prior arts already considered compressing ViTs, this paper stands out with its elegant formulation as a resource-constrained optimization problem, that is solved from end to end. It looks more synergistic and less ad-hoc.

I also like the rigorous optimization in this paper. Different from previous ad-hoc methods that need tune more hyperparameters, UVC can compress with one global budget control, and jointly coordinate different compression forms automatically under the hood.

Model and Algorithm:
Besides the common distillation loss, one of the authors’ main new ideas is to enforce mixed-level group sparsity: the head dimension level, the head number level, and the block number level. The authors point out the rationale as: when pruned to same ratios, finer-grained sparsity (i.e., pruning in smaller groups) is unfriendly to latency, while coarser-grained sparsity (i.e., pruning in larger groups) is unfriendly to accuracy. The mixed-level group sparsity can hence more flexibly trade-off between latency and accuracy.
Particularly, including layer-skipping as a compression means is novel, and to my best knowledge not considered by previous joint compression works yet. The authors also present good rationale why directly skipping layers in ViT might be acceptable: due to the observed layer collapse in many trained ViTs.
For the algorithm part: while I am familiar with primal-dual or ADMM, I got lost when reading the ~2 pages of optimization derivation in section 3. Specifically, while laying out all details, it is difficult to follow what main algorithm novelty the authors intend to claim: is just this formulation idea novel? Or the overall primal-dual algorithm being novel? Or the solution to some subproblem novel?

Experiment and Analysis:
One biggest issue I have had - I am not convinced by the comparison with the current SOTA method of patch slimming. It shows UVC can compress to higher ratios, but meanwhile suffer from notably more accuracy losses. If I just look at DeiT-Tiny PatchSlimming (0.7 FLOPs) versus UCV (0.6 FLOPs), I’m under the impression that UVC drops accuracy more quickly.

While the authors argued UVC can also be seamlessly extended to include token reduction, leaving this vaguely as “future work” cannot directly support the current work’s merit over prior arts. Could the authors report a more aligned apple-to-apple comparison with PatchSlimming?

More questions regarding experiments: Why not reporting DeiT-small? And why not more baselines for DeiT-Tiny with token? Also, the choice of target FLOPs looks a bit arbitrary to me, and different methods are not fully aligned. Could the authors clarify how they’re chosen, or whether they’re cherry picked?

Writing:
The paper’s writing is overall polished. BUT the algorithm part (page 5-6, Algorithm 1) is unnecessarily lengthy, I think the readability would be better if many here were moved to supplementary. Also, all tables in the paper are not numbered – very weird and hard to refer to.


**Summary Of The Paper:**

ViT compression is a new sub-field just being studied this year. With a few previous methods exploring pruning or data-dropping separately, this paper is the first to integrate multiple compression methods as one joint optimization.

**Summary Of The Review:**

This seems to be an overall well-executed paper, with reasonable amounts of novelty and potential. However, the authors need clarify on a few raised questions, on which my final rating will depend.

---

> ### Author Response · Authors · 2021-11-20
> **Response to Reviewer UdwX (Cons 1)**
>
> We are very glad you had a positive initial impression, and we provide pointwise responses for your concerns below.
>
> **[Cons1. Is just this formulation idea novel? Or the overall primal-dual algorithm being novel? Or the solution to some subproblem novel?]**
>
> To answer what makes you confused, primal-dual is not novel, as it is a classical optimization algorithm in the literature. However, the main novelty lies in the unified formulation. In this formulation, our main innovations include:
>
> - (1) the solid, mathematically well-grounded joint optimization form, quoting Reviewer **UdwX**’s comments: “the rigorous optimization in this paper. Different from previous ad-hoc methods that need tune more hyperparameters, UVC can compress with one global budget control, and jointly coordinate different compression forms automatically under the hood”, and **K4Gq** “the efforts to formulate and solve ViT compression as an end-to-end budget-constrained optimization is appreciated”;
> - (2) the novel mixed-level group sparsity formulated for ViT compression, and particularly (quoting Reviewer **UdwX**’s words) “ including layer-skipping as a compression means is novel, and to my best knowledge not considered by previous joint compression works yet.”, and (quoting **K4Gq**) “Directly skipping layers is a fresh idea as a compression means and seems particularly to fit ViT due to its homogenous block structure and common feature collapse. This makes a brand-new angle to compress ViTs.”

---

> ### Author Response · Authors · 2021-11-20
> **Response to Reviewer UdwX (Cons 2)**
>
> **[Cons2. UVC suffers from notably more accuracy losses? Why not report DeiT-small? And why not more baselines for DeiT-Tiny with tokens? ]**
>
> *<UVC suffers more accuracy losses?>*
>
> We respectfully argue our methods are comparable with Patch Slimming in terms of the efficiency and performance tradeoff. For example, we conduct extra experiments to nearly match with the FLOPs size that Patch Slimming showed in their paper. For UVC, a model with 53% FLOPs acquires 71.8% top-1 accuracy which is comparable to 72.0% acquired by Patch Slimming. This small performance difference may be caused by a negligible difference of FLOPs size between 54% (Patch Slimming) and 53% (UVC). Besides, note that our method works orthogonally to the token selection technique that Patch Slimming uses, UVC is flexible in integrating other compression techniques in the framework, as we discussed in Cons3.
>
> *<Why not DeiT-Small?>*
>
> Thanks for pointing this out. To address your concern, We also add new results of DeiT-Small in Table R2 below, where the comparison with **Patch Slimming** is included. Note that due to the limited rebuttal time window, the experiments are still on the run, and UVC is still evolving.
>
> **Table R1**
>
> |Method  	|         Acc  	| FLOPs(G) | Compression Ratio (%)|
> |:-:|:-:|:-:|:-:|
> |DeiT-Small 		|        79.8		|     	4.6    | 100|
> |SCOP[1] 			|    77.5 (-2.3) 	| 	2.6 	| 56.4|
> |PoWER[2] 		|    78.3 (-1.5) 	| 	2.7 	| 58.7|
> |HVT [3] 			|    78.0 (-1.8)  	| 	2.4 	| 52.2|
> |Patch Slimming [4] 	|    79.4 (-0.4)  	| 	2.6	| 56.5|
> |UVC (Ours)		|    79.44 (-0.36)| 	2.65 	| 57.61|
> |UVC (Ours)		|    78.82 (-0.98)| 	2.32 	| 50.41|
>
> As seen from Table R1, UVC already outperformed recent strong competitors such as HVT (published in ICCV’21) and others. The newly updated result has a comparable top-1 accuracy of 79.44% to Patch Slimming, while the compression ratio measured by FLOPs is similar to what Patch Slimming obtains. We post this intermediate - non-final and not conclusive - result here to ensure timely feedback, but will also update newer results once we have it. We publicly promise that DeiT-small compression results will be reported in the final draft.
>
> *<Why not more DeiT-Tiny with distillation token?>*
>
> We follow the setups in previous works [4,5,6] and thus adopt the token-free version of DeiT-Tiny. To bear with fair comparison, we align our main experimental setting with them on DeiT-Tiny/Small/Base without the distillation token. Additionally, to demonstrate the effectiveness of our method when the distillation token is enabled, we provide the results of our method for your reference as shown in Table 1, to excavate the limit of performance.
>
> [1] Yehui Tang, Yunhe Wang, Yixing Xu, Dacheng Tao, Chunjing XU, Chao Xu, and Chang Xu. Scop: Scientific control for reliable neural network pruning. In H. Larochelle, M. Ranzato, R. Hadsell, M. F. Balcan, and H. Lin, editors, Advances in Neural Information Processing Systems, volume 33, pages 10936–10947. Curran Associates, Inc., 2020.
>
> [2] Saurabh Goyal, Anamitra Roy Choudhury, Saurabh Raje, Venkatesan Chakaravarthy, Yogish Sabharwal, and Ashish Verma. Power-bert: Accelerating bert inference via progressive word-vector elimination. In International Conference on Machine Learning, pages 3690–3699. PMLR, 2020.
>
> [3] Pan, Zizheng and Zhuang, Bohan and Liu, Jing and He, Haoyu and Cai, Jianfei.Scalable Vision Transformers With Hierarchical Pooling. In Proceedings of the IEEE/CVF International Conference on Computer Vision (ICCV), pages 377-386, 2021.
>
> [4] Yehui Tang, Kai Han, Yunhe Wang, Chang Xu, Jianyuan Guo, Chao Xu, and Dacheng Tao. Patch slimming for efficient vision transformers. arXiv preprint arXiv:2106.02852, 2021.
>
> [5] Tianlong Chen, Yu Cheng, Zhe Gan, Lu Yuan, Lei Zhang, and Zhangyang Wang. Chasing sparsity in vision transformers: An end-to-end exploration. arXiv preprint arXiv:2106.04533, 2021b.
>
> [6] Mingjian Zhu, Kai Han, Yehui Tang, and Yunhe Wang. Visual transformer pruning. arXiv preprint arXiv:2104.08500, 2021.

---

> ### Author Response · Authors · 2021-11-20
> **Response to Reviewer UdwX (Cons 3&4&5)**
>
> **[Cons3. Report a more aligned apple-to-apple comparison with PatchSlimming]**
>
> It is worth mentioning that our proposed frameworks can integrate many other compression mechanisms into the same framework. Especially what the reviewer mentioned here with PatchSlimming, which is the token-level compression that can massively reduce the computation cost by calculating a smaller attention map. Since the code of Patch Slimming is provisionally not publicly available, we implemented a token selection technique using the code base of SViTE[1] (Specifically, SViTE+) into part of UVC, during the rebuttal window, and the model is currently running with promising intermedia results. We test this UVC-s (short for “UVC augmented by SVITE+ patch reduction“) on the DeiT-Tiny model and the result is listed in the following Table R2:
>
> **Table R2**
>
> |Method  	|         Acc  	| FLOPs(G) | Compression Ratio (%)|
> |:-:|:-:|:-:|:-:|
> |DeiT-Tiny |        72.2		|     	1.3    | 100|
> |UVC		|    71.3 (-0.9)  	| 	0.64 	| 49.23|
> |UVC-s	|    71.5 (-0.7) 	| 	0.65 	| 50.12|
>
> UVC-s framework can include Patch Slimming work into our primal-dual optimization, instead of deriving their original greedy method.  And UVC-s can perform better than UVC-only. This result proves that UVC and Patch Slimming are orthogonal to each other and adding it to our framework can further boost the performance.
>
> **[Cons4. Could the authors clarify how the FLOPs are chosen, or whether they’re cherry-picked?]**
>
> We want to clarify that the FLOPs are more determined by optimization. An important thing to note is that the resource cost measured by FLOPs is bounded by a global pre-selected resource budget, and the budget later serves as one of the constraints to be optimized in the primal-dual algorithm. Since there are other constraints too, this specific constraint might be ``slack” during optimization, and practically staying with some value close to the threshold after coordinating with other constraints.  The presented models with the target FLOPs are all the result of selecting 0.5 as the target resource budget.
>
>
> **[Cons5. The algorithm part is unnecessarily lengthy and the table is unmarked]**
>
> We tremendously appreciate your comments about the write-up. In the final version, we will reduce the algorithm part with only necessary details in around 1.2 pages; and the extra space will be devoted to displaying the Cons 1 - Cons 3 new results. All the tables will definitely be marked. For your reference, we upload a new version of the paper that has marked tables and the algorithm is moved to the appendix.
>
> [1] Tianlong Chen, Yu Cheng, Zhe Gan, Lu Yuan, Lei Zhang, and Zhangyang Wang. Chasing sparsity in vision transformers: An end-to-end exploration. arXiv preprint arXiv:2106.04533, 2021b.

---

> ### Comment · Reviewer_UdwX · 2021-11-26
> **Thanks for the response.**
>
> Thank the authors for providing many extra experimental results, including DeiT-Small, T2T, as well as the new integration of patch reduction into UVC. I'm now more convinced and will raise our score.

---

### Official Review · Reviewer_iRBq · 2021-11-02

**Correctness:** 2
**Technical Novelty And Significance:** 3
**Empirical Novelty And Significance:** 2
**Recommendation:** 5
**Confidence:** 5

**Main Review:**

Strengths

* Model compression is well studied in the fields of CNNs in CV and Transformer models in NLP. But such techniques still need to be adopted and verified in visual Transformers.

* The paragraphs in this paper are clear and easy to understand.

* Experiments in ImageNet with DeiT models prove the effectiveness of the proposed method. Compared with baseline models and recent Transformer compression methods, UVC can obtain larger compression rates, resulting in higher accuracies.

Weaknesses

* The novelty of the paper is quite limited. It is like a combination of existing methods, such as Slimming, BigNAS.

* The whole process of UVC is consists of compression training and post-training. I suspect that the comparison between different methods is not aligned, and post-training leads better performance.

* I suggest that UCV can be verified on more Transformer architectures, such as Swin/T2T/CvT/CeiT, showing the generalization ability.

* All tables are unmarked, which are hard to follow.



**Summary Of The Paper:**

This paper proposed a unified ViT compression framework that assembles pruning, layer skipping, and knowledge distillation as one. By adding the sparsity loss and the resources loss into the objective, ViT models can be stably trained at a high sparse ratio. By introducing the knowledge distillation loss, the compressed model can maintain performance with around 50% of the original FLOPs. Experimental results in ImageNet with DeiT models prove the effectiveness of the proposed method.

**Summary Of The Review:**

Overall, UVC has made a good attempt at model compression for visual Transformer models. But the novelty is limited. This method needs to provide a more rigorous comparison, and at the same time, it needs to be verified on more visual Transformer architectures to prove its effectiveness and generalization.

---

> ### Author Response · Authors · 2021-11-20
> **Response to Reviewer iRBq (Cons 1&2)**
>
> **[Cons1. Novelty limited?]**
>
> We respectfully disagree. The whole framework is clearly not a simple combination of different techniques. In the “new sub-field of ViT compression” (quoting Reviewer **UdwX**), our main innovations include:
>
> - (1) the solid, mathematically well-grounded joint optimization form, quoting **UdwX**: “the rigorous optimization in this paper... Different from previous ad-hoc methods that need tune more hyperparameters, UVC can compress with one global budget control, and jointly coordinate different compression forms automatically under the hood”, and **K4Gq** “the efforts to formulate and solve ViT compression as an end-to-end budget-constrained optimization is appreciated”;
>
> - (2) the novel mixed-level group sparsity formulated for ViT compression, and particularly (quoting **UdwX**) “ including layer-skipping as a compression means is novel, and to my best knowledge not considered by previous joint compression works yet.”, and (quoting **K4Gq**) “Directly skipping layers is a fresh idea as compression means and seems particular to fit ViT due to its homogenous block structure and common feature collapse. This makes a brand-new angle to compress ViTs.”
>
> You also mentioned two prior works which we fail to understand why they are very related. We respond below:
>
> * Slimming: We are not sure whether you wanted to refer to the work of PatchSlimming or Network Slimming
>     - (a) For PatchSlimming (PS), that work mainly focuses on compressing the layerwise token number to reduce the computational resource. While UVC mainly focuses on weight parameters and skip configuration, it is orthogonal to PS as we have emphasized in the paper, which is agreed by the other two reviewers. The token pruning strategy can also be integrated into the framework of UVC.
>     - (b) For NetworkSlimming (NS), it’s an effective channel pruning method implemented for CNN based on batch norm, but ViT has completely different building blocks, e.g., Self-Attentions (using layer norm) so we do not see how NS can be directly plugged in. Also, NS cannot achieve the mixed-level group sparsity in joint optimization, which is our well-recognized main contribution by other reviewers.
>
> * BigNAS is a Neural Architecture Search (NAS) method for CNNs, which really seems out of scope here. We do agree NAS and pruning have some similarities, but they are in general two fields and we belong to the latter. We conjecture you might misunderstand our skip connection and attention head search as similar to NAS. However, as we also explained above, ViT has different blocks from CNNs (see above); and our joint prime-dual optimization is a totally different method.
>
> **[Cons2. Post-training leads to better performance?]**
>
> We first clarify that there is not any “secret sauce post-training” in UVC; in fact, most existing compression-after-training methods need to go through pre-training-compression-retraining (the so-called post-training, or called fine-tuning), the last two steps of which may even need to be iterated. Re-training is crucial to restoring the accuracy loss due to removing weights and is used by prior ViT compression works such as PatchSlimming, IA-RED2, and VTP. By convention, they re-use the same (pre-)training recipe for the original target model. We followed this convention in all experiments when using different ViT models. Hence there is nothing ad-hoc of our compression process, and hence the comparison with prior ViT compression works is well aligned.

---

> ### Author Response · Authors · 2021-11-20
> **Response to Reviewer iRBq (Cons 3&4)**
>
> **[Cons3. More Transformer architectures]**
>
> Thanks for pointing this out. We have applied UVC on the backbone of T2T-ViT-14 to test the effectiveness of our proposals across different architectures. Since T2T-ViT-14 has much larger computational costs and is slower to train/run, during this limited rebuttal time and resource, as of now, the updated results on T2T-ViT-14 on diverse ratios are listed below:
>
> **Table R1**
>
> |Method  	|         Acc  	| FLOPs(G) | Compression Ratio (%)|
> |:-:|:-:|:-:|:-:|
> |T2T-ViT-14 		|        81.5		|     	4.8    | 100	|
> |PoWER[2] 		|    79.9(-1.6)	| 	3.5 	| 72.9|
> |UVC (high)		|    80.4 (-1.1) 	| 	2.90 	| 60.4|
> |UVC (medium)		|    79.6 (-1.9) 	| 	2.47 	| 51.5|
> |UVC (low)			|    78.9 (-2.6) 	| 	2.11 	| 44.0|
>
> As is shown in Table R1, UVC with 44% FLOPs achieves only a 2.6% accuracy drop. Meanwhile, UVC achieves a 1.9% accuracy drop with 51.5% of the original FLOPs. These two results are quite significant with a much larger compression ratio than PoWER performs. While with 60.4% FLOPs, UVC only suffers from a 1.1% accuracy drop, significantly outperforming PoWER with 72.9% FLOPs. This provides evidence that our method is independent of backbone and works well on both DeiT and T2T-ViT. We post this intermediate - non-final and not conclusive - result here to ensure timely feedback, but will also update newer results once we have it. We publicly promise that the T2T-ViT-14 compression results will be reported in the final draft.
>
>
> We also add new results of DeiT-Small in Table R2 below, where the comparison with **Patch Slimming** is included. Note that due to the limited rebuttal time window, the experiments are still on the run, and UVC is still evolving.
>
> **Table R2**
>
> |Method  	|         Acc  	| FLOPs(G) | Compression Ratio (%)|
> |:-:|:-:|:-:|:-:|
> |DeiT-Small 		|        79.8		|     	4.6    | 100|
> |SCOP[1] 			|    77.5 (-2.3) 	| 	2.6 	| 56.4|
> |PoWER[2] 		|    78.3 (-1.5) 	| 	2.7 	| 58.7|
> |HVT [3] 			|    78.0 (-1.8)  	| 	2.4 	| 52.2|
> |Patch Slimming [4] 	|    79.4 (-0.4)  	| 	2.6	| 56.5|
> |UVC (Ours)		|    79.44 (-0.36)| 	2.65 	| 57.61|
> |UVC (Ours)		|    78.82 (-0.98)| 	2.32 	| 50.41|
>
> As seen from Table R2, UVC already outperformed recent strong competitors such as HVT (published in ICCV’21) and others. The newly updated result has a comparable top-1 accuracy of 79.44% to Patch Slimming, while the compression ratio measured by FLOPs is comparable to what Patch Slimming obtains. We post this intermediate - non-final and not conclusive - result here to ensure timely feedback, but will also update newer results once we have it. We publicly promise that DeiT-small compression results will be reported in the final draft.
>
> **[Cons4. All tables are unmarked]**
>
> We tremendously appreciate your advice about unmarked tables,  and have uploaded a new pdf with marked tables.
>
> [1] Yehui Tang, Yunhe Wang, Yixing Xu, Dacheng Tao, Chunjing XU, Chao Xu, and Chang Xu. Scop: Scientific control for reliable neural network pruning. In H. Larochelle, M. Ranzato, R. Hadsell, M. F. Balcan, and H. Lin, editors, Advances in Neural Information Processing Systems, volume 33, pages 10936–10947. Curran Associates, Inc., 2020.
>
> [2] Saurabh Goyal, Anamitra Roy Choudhury, Saurabh Raje, Venkatesan Chakaravarthy, Yogish Sabharwal, and Ashish Verma. Power-bert: Accelerating bert inference via progressive word-vector elimination. In International Conference on Machine Learning, pages 3690–3699. PMLR, 2020.
>
> [3] Pan, Zizheng and Zhuang, Bohan and Liu, Jing and He, Haoyu and Cai, Jianfei.Scalable Vision Transformers With Hierarchical Pooling. In Proceedings of the IEEE/CVF International Conference on Computer Vision (ICCV), pages 377-386, 2021.
>
> [4] Yehui Tang, Kai Han, Yunhe Wang, Chang Xu, Jianyuan Guo, Chao Xu, and Dacheng Tao. Patch slimming for efficient vision transformers. arXiv preprint arXiv:2106.02852, 2021.

---

> ### Author Response · Authors · 2021-11-25
> **Response to Reviewer iRBq**
>
> Dear Reviewer iRBq,
>
> We thank the reviewer time for the review. To address your concerns, we provide detailed pointwise answers and we really hope to have a further discussion with reviewer iRBq to see if our response solves the concerns.
>
> We would sincerely appreciate it if reviewer iRBq could reply to the most important points in our rebuttal. For example, we provide newly added experimental results on T2T-ViT-14 and DeiT-Small and detailed explanation pointwise to your concern on the novelty of the work.
>
> We genuinely hope reviewer iRBq could kindly check our response. Thank you!
>
> Best wishes,
>
> Authors

---

> ### Author Response · Authors · 2021-11-28
> **Response to Reviewer iRBq**
>
> Dear Reviewer iRBq,
>
> We really appreciate your time and constructive reviews.
>
> We’d like to send you a little kind reminder that the discussion period is ending in three days. Detailed replies and new experiments are provided to specifically address your concerns. A chance for further discussion with you is precious to us to see if our responses solve your concerns.
>
> Given all the new experiments and replies, are you willing to reconsider your rating? Your support is very important to us and we greatly appreciate that!
>
> Best Regards,
>
> Authors

---

> ### Author Response · Authors · 2021-11-30
> **Sincerely expecting feedback from Reviewer iRBq**
>
> Dear Reviewer iRBq,
>
> We greatly appreciated your time and constructive reviews.
>
> We politely send you a kind reminder that the discussion period is ending **within 7 hrs**.
>
> So far, reviewers **UdwX** and **K4Gq** have already recognized our paper's merit and reached a consensus on an acceptance suggestion. It would be greatly appreciated if you could check the updated experimental results and detailed explanations that address your concerns so that you could also support our acceptance? It is crucial to have your support, and we will be immensely grateful for it.
>
> Last but not least,  if there are other clarifications or experiments we can provide, please do not hesitate to start a discussion with us. Thank you!
>
> Best Regards,
>
> Authors

---

### Author Response · Authors · 2021-11-29
**Summary of Updates (Many thanks to all reviewers and AC)**

Dear Reviewers and AC panel,

We are grateful for the valuable problem raised and the advice given by the reviewers that have helped revise our manuscript. It is glad to see that the merits of our work have been recognized by reviewers **UdwX** and **K4Gq**. Since the discussion period is ending in less than 2 days and we are yet to hear from reviewer **iRBq**, as we have provided detailed pointwise answers that could help to solve reviewer **iRBq**’s concerns and lead to a better understanding of the work. We are summarizing the major points about the paper (post-rebuttal) for a quick understanding of all:

**[Concern of novelty]** We clarify to our reviewers that the whole framework is not a simple combination of different techniques. Our main innovations include two aspects: (1) the solid, mathematically well-grounded joint optimization form that is recognized by Reviewer **UdwX**. (2) the novel mixed-level group sparsity formulated for ViT compression that is recognized by Reviewer **UdwX** and Reviewer **K4Gq**.

**[Concern of post-training]** We give evidence to prove the necessity of post-training in most compression procedures by referring to prior ViT compression algorithms such as PatchSlimming, IA-RED2, and VTP. As is also recognized in prior works, re-training is crucial to restoring the accuracy loss due to removing weights. Hence there is nothing ad-hoc about our compression process.

**[More transformer architectures]** We provide updated results of UVC on **DeiT-Small** and another architecture **T2T-ViT** that show great empirical novelty and significance. In Table R1 and R2 for Reviewer **iRBq**, UVC obtains good performance with a fairly low compression ratio. These results prove the effectiveness of UVC across different architectures.

**[Concern on the selection of FLOPs]** We clarify that the selection of FLOPs is more determined by the optimization process. The specific constraint of the optimization process may be practically staying with some value close to the threshold after coordinating with other constraints.

**[Sequential compression]** We show in Table R2 for reviewer **K4Gq** that sequentially applying these compression techniques is not optimal compared to joint optimization.

**[A few writing concerns]** We made changes in the manuscript with suggestions to mark the tables and to remove the lengthy algorithm. A new version of pdf is uploaded with the specified modifications.

We genuinely hope that reviewer **iRBq** could kindly check our response and have a further discussion to see whether the concerns have been well-addressed. It would be sincerely appreciated if reviewer **iRBq** could reply to our rebuttal. Thank you!

Best wishes,

Authors

---

### Decision · Program_Chairs · 2022-01-20

**Decision:**

Accept (Poster)

**Comment:**

This paper receives positive reviews. The authors provide additional results and justifications during the rebuttal phase. All reviewers find this paper interesting and the contributions are sufficient for this conference. The area chair agrees with the reviewers and recommends it be accepted for presentation.